

# The oldest record of Saurosphargiformes (Diapsida) from South China could fill an ecological gap in the Early Triassic biotic recovery

Long Cheng[1], Benjamin C. Moon[2], Chunbo Yan[1], Ryosuke Motani[3], Dayong Jiang[4], Zhihui An[1] and Zichen Fang[5]

[1] Hubei Key Laboratory of Paleontology and Geological Environment Evolution, Wuhan Center of China Geological Survey, Wuhan, P. R. China
[2] Palaeobiology Research Group, School of Earth Sciences, University of Bristol, Bristol, UK
[3] University of California Davis, Department of Earth and Planetary Sciences, Davis, California, United States of America
[4] Peking University, Department of Geology and Geological Museum, Beijing, P. R. China
[5] China University of Geoscience, Wuhan, P. R. China

Corresponding authors
Long Cheng, chengl@mail.cgs.gov.cn
Benjamin C. Moon,
benjamin.moon@bristol.ac.uk

## ABSTRACT

Diversification following the end-Permian mass extinction marks the initiation of Mesozoic reptile dominance and of modern marine ecosystems, yet major clades are best known from the Middle Triassic suggesting delayed recovery, while Early Triassic localities produce poorly preserved specimens or have restricted diversity. Here we describe *Pomolispondylus biani* gen. et sp. nov. from the Early Triassic Nanzhang-Yuan'an Fauna of China assigned to Saurosphargiformes tax. nov., a clade known only from the Middle Triassic or later, which includes Saurosphargidae, and likely is the sister taxon to Sauropterygia. *Pomolispondylus biani* is allied to Saurosphargidae by the extended transverse processes of dorsal vertebrae and a low, table-like dorsal surface on the neural spine; however, it does not have the typical extensive osteoderms. Rather an unusual tuberous texture on the dorsal neural spine and rudimentary ossifications lateral to the gastralia are observed. Discovery of *Pomolispondylus biani* extends the known range of Saurosphargiformes and increases the taxic and ecological diversity of the Nanzhang-Yuan'an Fauna. Its small size fills a different ecological niche with respect to previously found species, but the overall food web remains notably different in structure to Middle Triassic and later ecosystems, suggesting this fauna represents a transitional stage during recovery rather than its endpoint.

## INTRODUCTION

The origin and evolution of marine reptiles in the Early Triassic is the subject of intense study to elucidate the tempo and mode of biotic recovery following the end-Permian mass extinction, the largest of the Phanerozoic: (EPME; *Benton et al., 2013*; *Ezcurra & Butler, 2018*; *Alvarez et al., 2019*). While several sites preserve Early Triassic strata globally,

only those in China have produced substantially complete and articulated remains of marine reptiles to date (*Cheng et al., 2015*; *Ezcurra & Butler, 2018*; *Li & Liu, 2020*). This contrasts with terrestrial vertebrate ecosystems of this time, from which numerous specimens, albeit of a few common "disaster taxa", have been found, *e.g., Lystrosaurus* (*Sahney & Benton, 2008*; *Romano et al., 2020*). Key innovations in the Early Triassic oceans include the incursion of several disparate marine reptile groups (*Chen et al., 2014c*; *Scheyer et al., 2017*), establishing reptiles in the top tiers of marine ecosystems (*Scheyer et al., 2014*; *Li & Liu, 2020*). Despite rapid diversification taxonomically, morphologically, and ecologically (*Moon & Stubbs, 2020*; *Reeves et al., 2021*), it was not until the Middle Triassic that clades that will dominate the rest of the Triassic (Ichthyopterygia, Sauropterygia, Thalattosauria) become established and built more complex ecosystems (*Fröbisch et al., 2013*; *Benton et al., 2013*; *Sander et al., 2021*). The question of how quickly and smoothly recovery occurred and what roles biotic and abiotic interactions play remains. Establishing the taxic and ecological diversity of Early Triassic ecosystems remains integral to understanding recovery patterns following the EPME.

Saurosphargidae are an enigmatic family of marine reptiles primarily known from the Middle Triassic of Europe and China. The first specimens were identified from the Alps (*Frech, 1903*; *von Huene, 1936*; *Nosotti & Rieppel, 2003*), but recent specimens predominantly from China have increased the diversity of Saurosphargidae and provided new hypotheses on its origin and relationships to other major groups. Saurosphargidae was erected by *Li et al. (2011)* to include *Saurosphargis volzi* and *Sinosaurosphargis yunguiensis* (*von Huene, 1936*; *Nosotti & Rieppel, 2003*); *Largocephalosaurus polycarpon* and *L. qianensis* were added later (*Cheng et al., 2012*; *Li et al., 2014*). Saurosphargidae are characterized by elongate transverse processes on the neural spine of dorsal vertebrae, a transverse expansion and pachyostosis of the dorsal ribs, and by the presence of osteoderms (*Li et al., 2011*). *Eusaurosphargis dalsassoi* and *Helveticosaurus zollingeri*, both from the Middle Triassic of Switzerland and Italy, are possibly terrestrial animals that share some features with Saurosphargidae (*Peyer, 1955*; *Rieppel, 1989*; *Scheyer et al., 2017*), however, these two taxa have been excluded by recent phylogenetic analyses (*Li et al., 2011*, *2014*; *Chen et al., 2014c*). Similarly, the presence of osteoderms and the rib morphology has led to comparisons with the early turtles *Odontochelys* and *Eorhynchochelys* (*Li et al., 2008*, *2018*), but the formation of the plastron in these taxa differs enough to separate them (*Li et al., 2014*).

In recent years, Wuhan Centre of China Geological Survey (WGSC) has led several field excavations of the Lower Triassic Jialingjiang Formation between Nanzhang and Yuan'an counties, Hubei, China, collecting abundant marine reptiles from the Nanzhang-Yuan'an Fauna (NYF; *Cheng et al., 2015*). The NYF is mainly represented by primitive marine reptiles with rare invertebrates. The marine reptiles in this fauna include hupehsuchians (*Hupehsuchus* (*Carroll & Dong, 1991*), *Parahupehsuchus* (*Chen et al., 2014b*), *Nanchangosaurus* (*Wu et al., 2003*; *Chen et al., 2014c*), *Eohupehsuchus* (*Chen et al., 2014a*), *Eretmorhipis* (*Chen et al., 2015*; *Cheng et al., 2019*)), sauropterygians (*Hanosaurus* (*Young & Dong, 1972*; *Rieppel, 1998*), *Keichousaurus* (*Young, 1965*), *Lariosaurus* (*Chen et al., 2016*; *Li & Liu, 2020*)), and one ichthyosauriform (*Chaohusaurus*

*zhangjiawanensis* (*Chen et al., 2013*)) (*Cheng et al., 2015*). Here, we describe a new marine reptile genus and species from the NYF. Although it is only represented by postcranial elements, it shares critical characters with Saurosphargidae suggesting these evolved earlier than previously thought.

## GEOLOGICAL SETTING

The South China Block is composed of the Yangtze Craton and the South China Fold Belt that together formed a stable continent from the late Proterozoic or early Paleozoic (*Shen et al., 2006*) into the early Mesozoic. During this time deposition was characterized by shallow-water carbonates. In the Early Triassic, the Yangtze Carbonate Platform extended approximately 1,200 km east to west and 500 km north to south, between the deeper-water Guizhou-Guangxi-Hunan (also Nanpanjiang) Basin to the south and the Yangtze Basin to the east (*Benton et al., 2013*; *Feng et al., 2015*). This expanse of shallow water was interrupted only occasionally by small banks and potential exposure that formed localized lagoonal areas (Fig. 1B). Sources of clastic material into the basin were dominated by the Cathaysia Land to the south-east and Kangdian Land to the west. These strata yield several marine reptile faunas through the Early to Late Triassic in the Yangtze Craton (*Benton et al., 2013*), including the Early Triassic NYF.

The NYF is present through the upper part of Member II of the Jialingjiang Formation in western Hubei Province, China (Figs. 1C, 1D). Earlier references indicated that these fossils were from the upper part of Member III, however, the local stratigraphy of the Jialingjiang Formation has recently been revised (*Yan et al., 2021*). These strata were deposited during the Spathian (Olenekian, Early Triassic) towards the northern margin of the preserved Yangtze Carbonate Platform (Fig. 1B). Despite the extent of this shallow-water platform, occurrences of the NYF are limited to a small region north of Yichang, from six localities (Fig. 1A): Gujing, Songshugou, Baihechuan, Xunjianzhen, Zhangjiawan, and Yingzishan (*Cheng et al., 2015*; *Qiao, Iijima & Liu, 2020*). These localities were all deposited in lagoonal environments within the broader carbonate platform, although its extent is uncertain, becoming occasionally more restricted (Fig. 1B; *Yan et al., 2021*). Members II and III of the Jialingjiang Formation correlate approximately to the Nanlinghu Formation found in the eastern Yangtze Platform, which preserves the Chaohu Fauna at Majiashan, Anhui Province (*Benton et al., 2013*). Despite generic overlap in the marine tetrapod fauna (*e.g., Chaohusaurus*; *Huang et al., 2019*), no species have been identified at both localities, and Hupehsuchia are unique to the NYF.

## MATERIALS AND METHODS

### Materials

The new taxon is described from a single specimen (WGSC V 1701) in two parts housed at WGSC. The part specimen (WGSC V 1701-1) preserves the articulated trunk portion of the body, prepared in dorsal view, whereas the counterpart (WGSC V 1701-2) mainly shows skeletal impressions and some bone fragments posteriorly. This specimen was collected from the Lower Triassic Member II of the Jialingjiang Formation at Songshugou Quarry near Xuanjianzhen (Xuanjian Village), Nanzhang County, Hubei Province

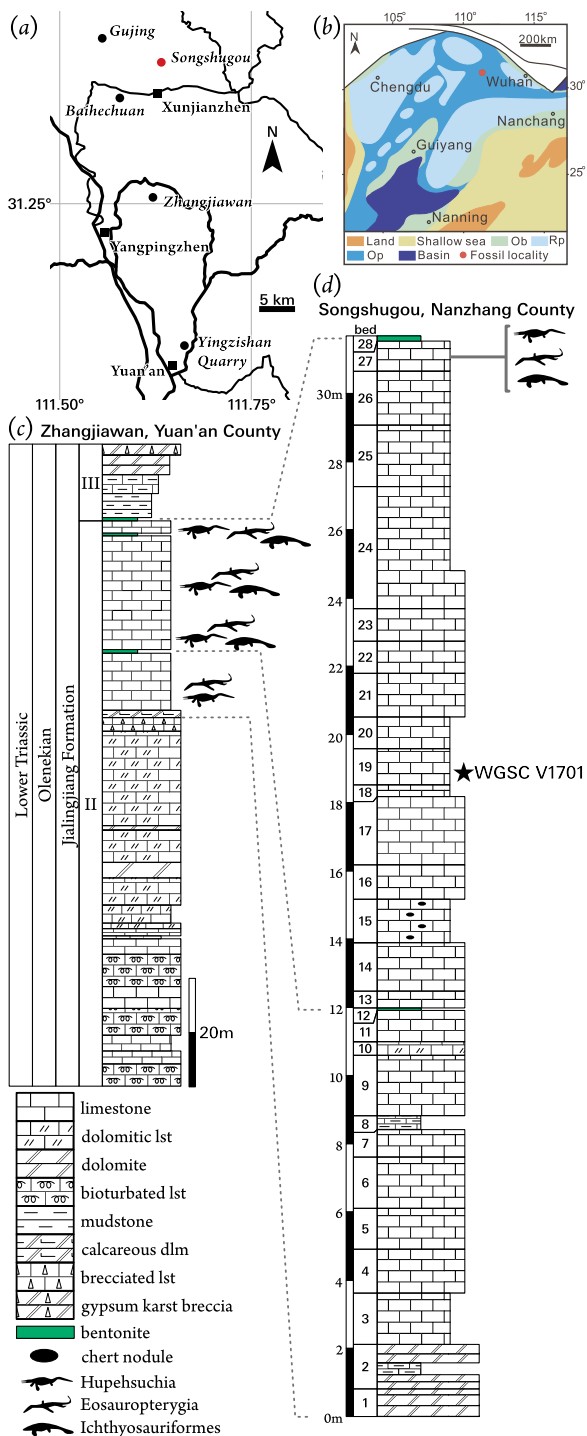

**Figure 1 Locality information for *Pomolispondylus biani* get. et sp. nov. holotype specimen (WGSC V1701).** (A) Location of Songshugou Quarry, Nanzhang County, Hubei. China. (B) Palaeogeographical reconstruction of the South China platform during the Early Triassic after *Yan et al. (2021)*. (C, D) Stratigraphical sections and correlation between Zhangjiawan Quarry, Yuan'an County (C) and the holotype locality at Songshugou Quarry, Nanzhang County (D). *Pomolispondylus biani* is found lower in the Songshugou section than other marine tetrapods. Abbreviations: Ob, oolitic beach; Op, open carbonate platform; Rp, restricted carbonate platform.
(Figs. 1A, 1D); WGSC V 1701 was found lower in the sequence than other layers that yield fossil marine reptiles at Songshugou Quarry (Fig. 1D).

## Methods

We added WGSC V 1701 as a new operational taxonomic unit (OTU) to the cladistic matrix of *Li et al. (2014)* (Analysis 1) that focused on the phylogenetic relationships of Saurosphargidae to Sauropterygia and other diapsids, and to the matrix of *Chen et al. (2014c)* as modified most recently by *Scheyer et al. (2017)* used to assess the relationships of early Mesozoic marine tetrapods. Following *Chen et al. (2014c)*, we run two analyses using this second matrix: (Analysis 2) incorporating all characters, and (Analysis 3) modifying marine-related characters to uncertainty (see Supplemental Information). We also added OTUs representing Hupehsuchia and *Hanosaurus* to the matrix of *Li et al. (2014)* and modified two characters (see Supplemental Information). These three analyses provide complementary results for detailed relationships of saurosphargids (*Li et al., 2014*) and for broader diapsid and marine tetrapod relationships using a recent data set (*Chen et al., 2014c*; *Scheyer et al., 2017*). All analyses were implemented using maximum parsimony in TNT version 1.5 (*Goloboff & Catalano, 2016*), and Bayesian inference in MrBayes version 3.2.7a (*Ronquist et al., 2012*) using gamma- and log-normal-distributed rates models (see Supplemental Material). Convergence of the analyses in MrBayes was checked using estimated sample size (ESS) >200 and plotting traces of the MCMC output in R version 4.1.0 (*R Core Team, 2021*) and package CODA version 0.19.4 (*Plummer et al., 2006*). Model comparison used a stepping-stone analysis implemented in MrBayes (*Xie et al., 2011*) and comparisons of Bayes factors (*Kass & Raftery, 1995*).

The electronic version of this article in Portable Document Format (PDF) will represent a published work according to the International Commission on Zoological Nomenclature (ICZN), and hence the new names contained in the electronic version are effectively published under that Code from the electronic edition alone. This published work and the nomenclatural acts it contains have been registered in ZooBank, the online registration system for the ICZN. The ZooBank LSIDs (Life Science Identifiers) can be resolved, and the associated information viewed through any standard web browser, by appending the LSID to the prefix http://zoobank.org/. The LSID for this publication is: urn:lsid:zoobank. org:pub:2EF3BA4D-B401-4B86-88C9-9320466A1ACD. The online version of this work is archived and available from the following digital repositories: PeerJ, PubMed Central SCIE and CLOCKSS.

## SYSTEMATIC PALAEONTOLOGY

**Diapsida** *Osborn, 1903* (*Gauthier & de Queiroz, 2020*)

### Saurosphargiformes tax. nov., new clade name

**Phylogenetic definition:** The maximum clade including *Saurosphargis volzi von Huene, 1936*, but excluding *Placodus gigas Agassiz, 1839*, *Plesiosaurus dolichodeirus Conybeare, 1824*, and *Hupehsuchus nanchangensis Young & Dong, 1972*. **Abbreviated definition:** max ∇ (*Saurosphargis volzi von Huene, 1936*) | ~ (*Placodus gigas Agassiz, 1839* &

*Plesiosaurus dolichodeirus Conybeare, 1824* & *Hupehsuchus nanchangensis Young & Dong, 1972*).

**Discussion:** Diagnostic characters for this new clade are taken from the phylogenetic analyses presented below as well as comparing shared (non-phylogenetic) morphology with the diagnosis of Saurosphargidae presented by *Li et al. (2014)*; this is amended below.

**Diagnosis:** Aquatic diapsids characterized by the following combination of characters: (1) low neural spine of dorsal vertebrae with table-like top; (2) presence of dorsal osteoderms; (3) transverse processes of dorsal vertebrae distinctly long; (4) anterior ribs developing facets for ossicles dorsally; (5) gastralia elongate and flattened; (6) lateralmost elements of gastral ribs broadened and contacting each other; (7) humerus not expanded at both ends.

### Saurosphargidae Li et al., 2011
**Phylogenetic definition:** The minimum clade including *Saurosphargis volzi von Huene, 1936*, *Largocephalosaurus polycarpon Cheng et al., 2012*, and *Sinosaurosphargis yunguiensis Li et al., 2011*, and all its descendants. **Abbreviated definition:** min ∇ (*Saurosphargis volzi von Huene, 1936* & *Largocephalosaurus polycarpon Cheng et al., 2012* & *Sinosaurosphargis yunguiensis Li et al., 2011*).

**Discussion:** A phylogenetic definition for Saurosphargidae has not previously been set down; the name is based on inclusion of the type species *Saurosphargis volzi* (*Li et al., 2011*), and diagnosed based on *Largocephalosaurus* and *Sinosaurosphargis* (*Li et al., 2014*). We propose a minimum (node-based) clade definition matching the indications of Saurosphargidae in previous phylogenetic hypotheses, differentiating Middle Triassic Saurosphargidae from more plesiomorphic Saurosphargiformes. The diagnosis below is modified from *Li et al. (2014)* based on our phylogenetic analyses and comparing shared characters with the more inclusive Saurosphargiformes.

**Diagnosis:** Modified from *Li et al. (2014)*: (1) dorsal ribs forming a closed basket; (2) presence of large dorsal osteoderms [modified]; (3) external naris retracted; (4) median elements of gastral ribs with a two-pronged lateral process on one side; (5) supratemporal extensively contacting quadrate shaft; (6) posterior margin of skull roof deeply emarginated; (7) jugal-squamosal contact; (8) presence of ectopterygoid; (9) presence of interpterygoid vacuity and open braincase-palatal articulation; (10) leaf-shaped tooth crown with convex labial surface and concave lingual surface; (11) tip of neural spines covered by osteoderms [modified]; (12) large interclavicle boomerang-like or atypical T-shaped, with a small and sharp posterior process; (13) nine carpals; (14) four tarsals; (15) pachyostosis of dorsal ribs [new]; (16) distal end of transverse process distinctly thickened [new]; (17) deltopectoral crest absent [new]; (18) median gastral rib element two-pronged with lateral process on one side [new].

### *Pomolispondylus biani* gen. et. sp. nov.
**LSID:** zoobank.org:act:42A625CB-EE27-4432-8606-7DD0ADC760D4

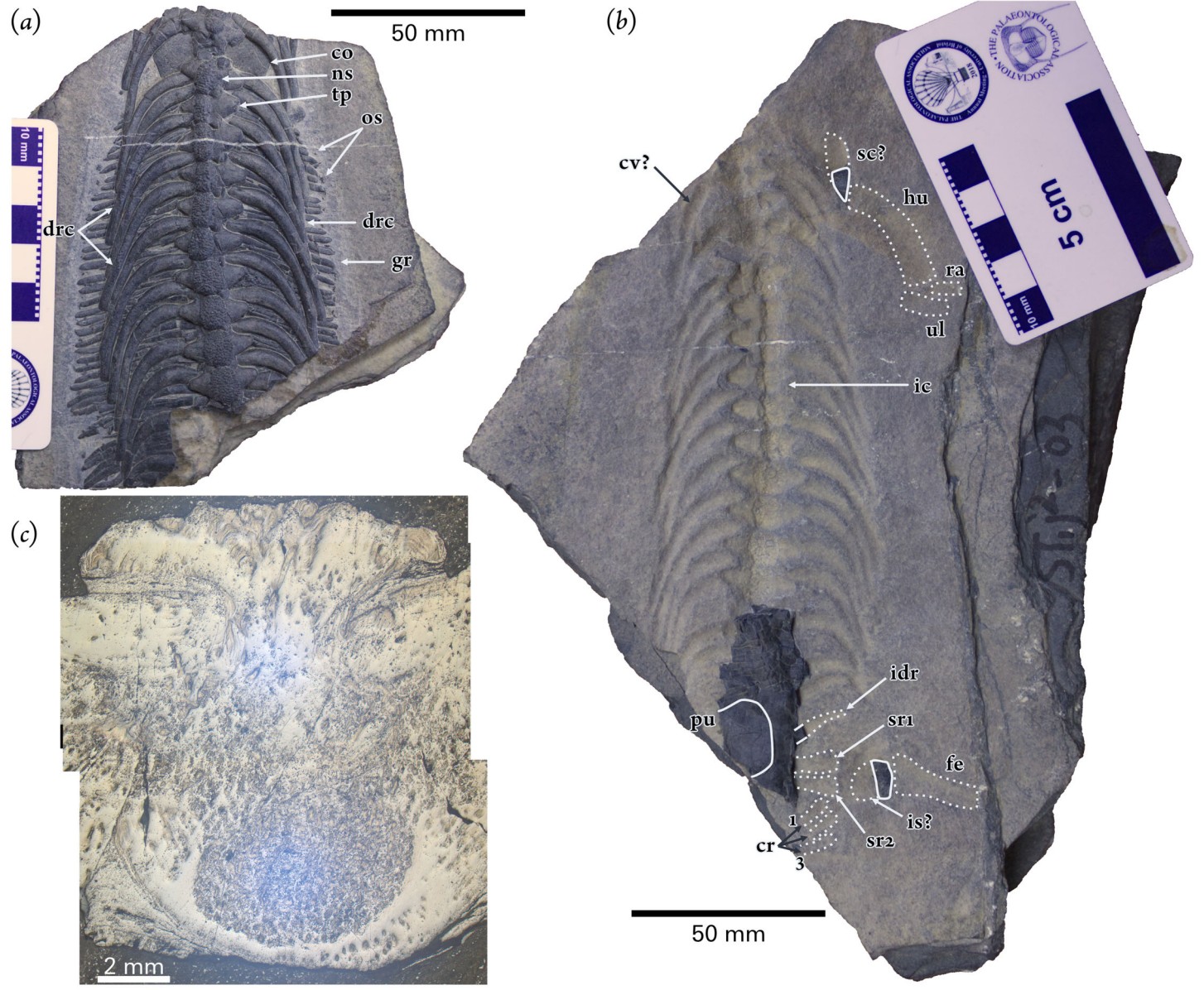

**Figure 2 Holotype specimen of *Pomolispondylus biani* gen. et sp. nov. (WGSC V1701).** (A) Part specimen (WGSC V1701-1) in dorsal view. (B) Counterpart specimen (WGSC V1701-2) in ventral view. Both images are shown to the same scale with anterior to the top of the image. (C) Transverse thin section through a vertebra. Abbreviations: co, coracoid; cr, impression of caudal ribs (1 and 3 are numbered); dr, dorsal rib; drc, dorsal rib crest; fe, impression of femur; gr, gastralia; hu, impression of humerus; ic, impression of straight-faced centrum; idr, impression of dorsal rib; is, impression of ischium; los, lateral ossification; ns, neural spine; pu, pubis; ra, impression of radius; sc, scapula; sr1/2, impressions of sacral ribs; tp, transverse process; ul, impression of ulna.

**Etymology:** The generic name *Pomolispondylus* is from the Greek 'πόμολα σπόνδυλο' (pómola spóndylo) meaning 'knobbly vertebra', referencing the morphology of the neural spines; the specific name *biani* honors Bian He, a famous Chinese historical figure from the locality.

**Holotype:** Part and counterparts: WGSC V 1701-1 and V 1701-2 (Figs. 2, 3). When WGSC V 1701 was removed from the matrix the rock split into two parts along the fossil layer and
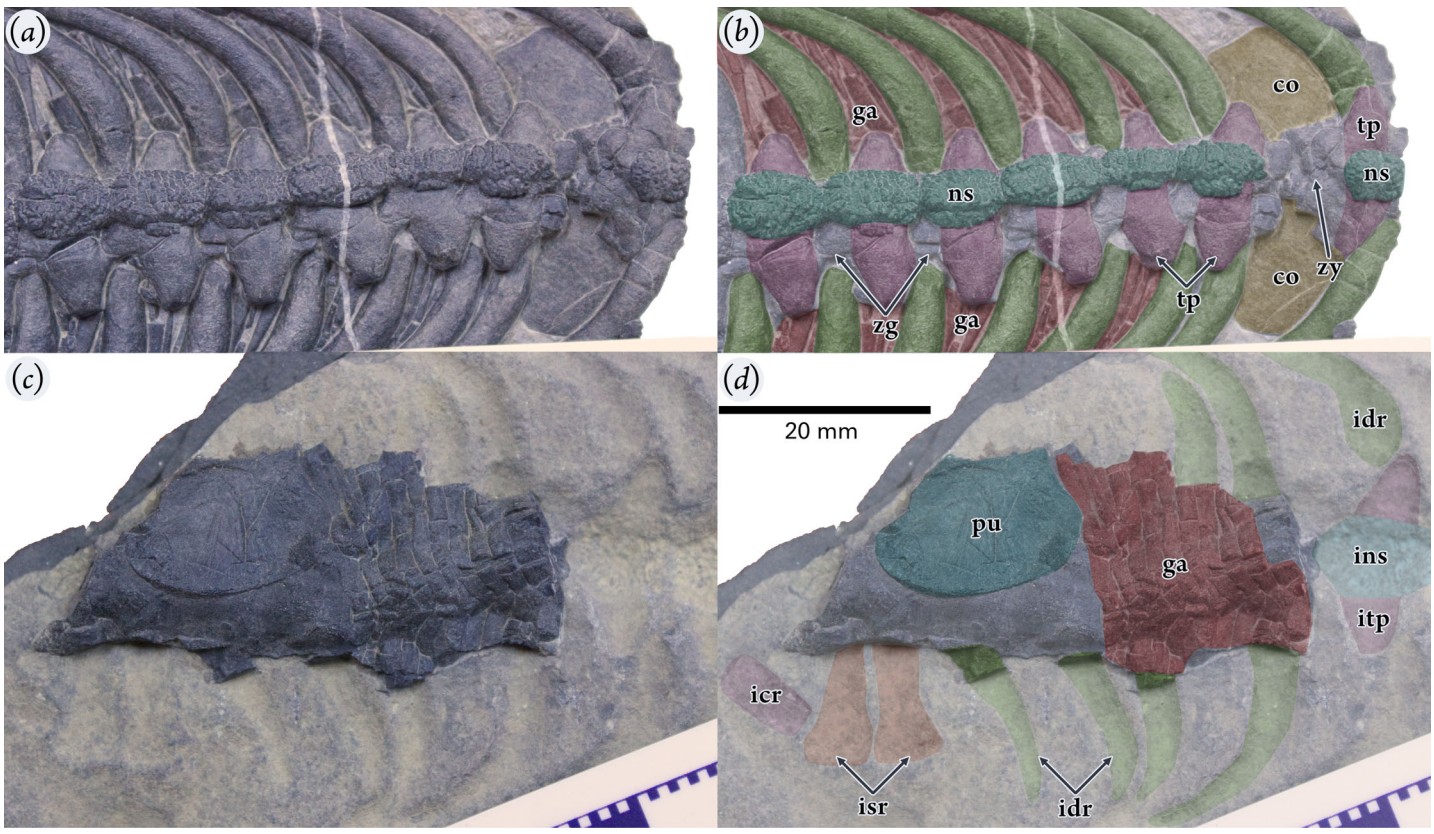

**Figure 3  Holotype specimen of *Pomolispondylus biani* gen. et sp. nov. (WGSC V1701).** (A, B) Oblique dorsal-right lateral view of the anterior of the part specimen (WGSC V1701-1) (A) with interpretation (B). (C, D) Oblique ventral-left lateral view of the posterior counterpart specimen (WGSC V1701-2) (C) with interpretation (D). All images are shown at the same scale with anterior to the right of the image. Abbreviations: co, coracoid; ga, gastralia; icr, impression of first caudal rib; idr, impression of dorsal rib; ins, impression of dorsal neural spine; isr, impression of sacral rib; itp, impression of transverse process; ns, neural spine; pu, pubis; tp, transverse process; zg, zygapophyses; zy, zygosphene-zygantrum articulation.

some elements were broken. Most elements are present on the part specimen (WGSC V 1701-1; Fig. 2A), while four posterior dorsal and two sacral vertebrae are on the counterpart with impressions of the neural spines, ribs, and fore and hind limbs of the right side (WGSC V 1701-2; Fig. 2B). The length of WGSC V 1701-1 is 128.54 mm, and the counterpart WGSC V 1701-2 is 200.8 mm long.

**Locality and Horizon:** Member II of the Jialingjiang Formation, Spathian (Olenekian), Early Triassic; Songshugou Quarry, Xuanjianzhen, Nanzhang County, Hubei Province, China. The definition of the Jialingjiang Formation was revised and three members were recognized based on geological mapping in Nanzhang and Yuan'an counties recently (*Yan et al., 2021*). The NYF is found in the top layers of Member II in the Jialingjiang Formation (Fig. 1).

**Diagnosis:** Saurosphargiformes with the following unique combination of characters: at least 18 dorsal vertebrae, two sacral vertebrae; neural spine heavily rugose dorsally; vertebral centrum with parallel lateral surfaces; proximal end of dorsal ribs slightly

**Table 1 Estimated presacral length of *Pomolispondylus biani* (WGSC V 1701).** Comparisons of humeral lengths to other Early–Middle Triassic Sauropterygia and Saurosphargidae.

| Taxon | Specimen | Length (mm) | Scaling factor | Total presacral length (mm) | Estimated presacral length (mm) of *Strumospina biani* |
|---|---|---|---|---|---|
| *Strumospina biani* | | 31.5 | | | |
| *Largocephalosaurus qianensis* | IVPP V 15638 | 164 | 5.21 | 800 | 154 |
| | GMPKU-P-1532-B (B) | 184 | 5.84 | 1,175 | 201 |
| *Sinosaurosphargis yuguiensis* | ZMNH M 8797 | 100 | 3.17 | 470 (carapace length) | 148 |
| *Eusaurosphargis dalsassoi* | PIMUZ A/III 4380 | 15.9 | 0.505 | 106 | 210 |
| *Dianmeisaurus gracilis* | IVPP V 18630 | 18.0 | 0.571 | 188 | 329 |
| *Diandongosaurus acutidentatus* | NMNS00093-F034398 | 20.0 | 0.635 | 170 | 268 |
| | | | | Mean length estimate (mm) | 218 |

pachyostotic; lateral margin of distal part of dorsal rib developing oval facet; dense, sheet-like gastralia, with four grouped elements corresponding to one centrum; a small ossicle attaching the distal end of the lateral element of gastralia; humerus curved more strongly than straight femur; medial margin of pubis rounded.

## DESCRIPTION

### Size

As preserved the specimen WGSC V 1701 measures 200.8 mm on the counterpart specimen (WGSC V 1701-2), while the dorsal portion is around 150 mm long. This size is comparable to that of *Keichousaurus yuananensis* from the same formation (dorsal length 145 mm in IVPP V2799; *Young, 1965*), and to small Middle Triassic pachypleurosaurs like *Dianopachysaurus dingi* (*Liu et al., 2011*) and *Dianmeisaurus gracilis* (trunk length 98 mm in IVPP V 18630; *Shang & Li, 2015*). The humerus impression indicates that it was 31.5 mm in WGSC V 1701. This is notably smaller than the humerus of *Lariosaurus sanxiaensis* (58.02 mm in HFUT YZS-16-01; *Li & Liu, 2020*) and *Hanosaurus hupehensis* (by way of comparison: the femur is 43.2 mm in IVPP V3231; *Young & Dong, 1972*; *Rieppel, 1998*). WGSC V 1701 is also smaller than Hupehsuchia from the same locality: the smallest *Eretmorhipis carrolldongi* has a dorsal length 255 mm (IVPP V4070; *Chen et al., 2015*) and *Eohupehsuchus brevicollis* is incomplete at 236 mm (WGSC 26003; *Chen et al., 2014b*), up to *Parahupehsuchus longus* with a dorsal length of 500 mm (WGSC 26005; *Chen et al., 2014a*). Estimates of presacral length vary given the different morphological proportions of compared taxa, but suggest little more than 200 mm, and likely smaller than 329 mm (Table 1).

### Ontogenetic assessment

Maturity of the specimen is determined from comparisons with closely related Sauropterygia and modern lepidosaurs and archosaurs. This is made problematic due to the focus on cranial determinants that aren't preserved in the material described herein. However, we identified the following characters that support a mature determination: neural spines fused to the vertebral centrum (*Klein, 2012*; *Griffin et al., 2021*); ossified

zygantrum-zygosphene articulation, long bone epiphyses, and ossified ulnare (*Lin & Rieppel, 1998*; *Griffin et al., 2021*); humerus as robust as the femur (*Lin & Rieppel, 1998*).

## Centra & Ribs

In WGSC V1701-1, there are 11 dorsal centra preserved that are covered by the neural arches dorsally, and six centra in WGSC V1701-2, which are obscured by the gastralia and pelvic elements. Adding two impressions of anterior dorsal vertebrae gives a total of around 20 presacrals (Fig. 2). The vertebral centra are oblong in dorsal view with straight lateral faces to the centra, unlike the concave shape seen in later Sauropterygia (*e.g.*, *Dianmeisaurus* (*Shang, Li & Wu, 2017*) and *Keichousaurus* (*Lin & Rieppel, 1998*)) and in *Largocephalosaurus* (*Li et al., 2014*). The anteriormost rib impression does not join to an impression of a vertebra, but more posterior elements are generally well articulated. The first rib impression is much shorter than the posterior neighboring impressions as in *Largocephalosaurus* (Fig. 2B; *Li et al., 2014*), suggesting that the first rib impression represents the last cervical rib. The shape and length of the second rib impression is like the posterior neighboring ones. The first partially preserved rib anteriorly and first completely preserved rib correspond to the second and third impressions, respectively. So, we interpret the partially preserved rib to be the first dorsal rib. Based on this interpretation, there are 18 dorsal and two sacral vertebrae present in both parts of the specimen; this is much fewer than in *Largocephalosaurus* (24 dorsal, two sacral centra; *Li et al., 2014*), *Lariosaurus sanxiaensis* (~26 dorsal; *Li & Liu, 2020*), and *Anarosaurus* (25 dorsals; *Klein, 2012*), but similar to the pachypleurosaurs *Keichousaurus yuananensis* (19–20 dorsals; *Young, 1965*) also from the Early Triassic, and *Dianopachysaurus* (19 dorsals, three sacrals; *Liu et al., 2011*), *Diandongosaurus* (18 dorsals, three sacrals; *Sato et al., 2014*), and *Dianmeisaurus* (18 dorsals, four sacrals; *Shang & Li, 2015*) from the Middle Triassic.

## Dorsal Vertebrae

The transverse processes of the dorsal vertebrae are elongate: about twice the lateral width of the centrum (*e.g.*, 9.06 *vs* 17.22 mm in the anteriormost preserved vertebra), or more than twice the width across the prezygapophyses (*e.g.*, 8.71 *vs* 18.52 mm in the eighth preserved vertebra). These are longer than in *Lariosaurus sanxiaensis* (*Li & Liu, 2020*) and in later small pachypleurosaurs (*e.g.*, *Dianmeisaurus* (*Shang, Li & Wu, 2017*), *Diandongosaurus* (*Liu et al., 2015*)), but shorter than in Saurosphargidae. Laterally the transverse processes narrow to a rhombus-like shape in dorsal view (Figs. 2B, 3A, 3B), rather than the more narrow, elongate forms that are found in *Saurosphargis* and *Largocephalosaurus* (*Nosotti & Rieppel, 2003*; *Li et al., 2014*). It appears to be rounded in cross section, but the specimen has been dorsoventrally compressed, so the original shape cannot be determined. The zygantrum-zygosphene articulation, as preserved, is present above and between the zygapophyses, as shown between dorsals four and five (Figs. 3A, 3B).

The neural spines of the new taxon are unusual. Each neural spine is very low (just exceeding the height of the neural arch; Figs. 2C, 3A, 3B, 4A), like in *Lariosaurus sanxiaenesis*, *Keichousaurus yuananensis*, and later pachypleurosaurs (Fig. 4D; *Young,*

*1965*; *Shang, Li & Wu, 2017*; *Li & Liu, 2020*), and does not extend dorsally as in *Sinosaurosphargis* (Fig. 4F; *Li et al., 2011*). The neural spine then expands horizontally near the neural arch, particularly anteroposteriorly (Figs. 3A, 3C, 4A), so that the neighboring two neural spines contact each other on their anterior and posterior margins (Figs. 2A, 3A, 3B). However, the neural spine differs from *Lariosaurus sanxiaenesis*, *Keichousaurus yuananensis*, *Hanosaurus hupehensis*, and pachypleurosaurs by quickly expanding posteriorly and becoming laterally broader towards its posterior (Figs. 2A, 3A, 3B, 4A, 4D, 4G; *Young, 1965*; *Rieppel, 1998*). Additionally, the size of the neural spines increases posteriorly along the vertebral column (Figs. 2A, 3A). The length/width ratios of neural spines 10 to 13 are 8.82/8.23, 8.68/7.27, 9.15/7.78, 9.32/7.62, respectively (Figs. 2A, 2C).

Each dorsal neural spine forms an elliptical table (Figs. 2C, 3A, 4A), as in *Largocephalosaurus*, *Eusaurosphargis*, Placodontoidea, and *Eorhynchochelys* (*Nosotti & Rieppel, 2003*; *Li et al., 2014*, *2018*; *Cheng et al., 2015*; *Scheyer et al., 2017*). However, the surface of this table in the new taxon is slightly convex dorsally, with abundant tuber-like ornaments in the surface, whereas the dorsal neural spine in *Largocephalosaurus*, *Eusaurosphargis*, and Placodontoidea is flattened or slightly concave (Figs. 3A, 3B, 4A, 4C, 4E). The tubers in the center of the neural spine's table are distinctly larger than the surrounding ones and sub-circular, while the surrounding tubers are elongated radially (Figs. 3A, 3B). In *Largocephalosaurus*, *Eusaurosphargis*, Placodontoidea, and *Eorhynchochelys*, the dorsal neural spines are covered by at least one layer of osteoderms. It is difficult to identify the shape in *Sinosaurosphargis*, because its body trunk is covered by small dense osteoderms (*Li et al., 2011*). However, there are no osteoderms preserved near the vertebral column in this specimen, indicating that the table of the neural spine was possibly covered by soft tissue.

## Dorsal Ribs

The dorsal ribs are single-headed (shown at their contact with the lateral centra and on anterior broken surfaces of the part specimen), as in *Lariosaurus sanxiaensis*, *Majiashanosaurus discocoracoideus*, and Middle Triassic pachypleurosaurs (*Jiang et al., 2014*; *Li & Liu, 2020*; *Liu et al., 2021*), and apparently articulate with the vertebral centra ventral to the transverse process (Figs. 2A, 3A, 3B, 4A); however, compaction of the specimen has made the positions of the rib uneven. Proximally, the ribs are somewhat thickened for approximately one-quarter of their length and narrow only slightly to the rather stout, rounded distal end of the rib. In cross section, the ribs are rounded and sub-circular.

The anterodorsal margin of the rib develops a low crest near the distal end in each of dorsal ribs 1 to 15. The crest forms an elliptical facet that extends distally along the shaft of the rib with a flat, smoothed surface (Figs. 2A, 4B). This is different from *Saurosphargis*, *Largocephalosaurus*, and *Eusaurosphargis* in which the dorsal rib develops an uncinate process at the shoulder region (Fig. 4E; *Nosotti & Rieppel, 2003*; *Li et al., 2014*), however, no uncinate process is present in *Sinosaurosphargis* (Fig. 4F; *Li et al., 2011*). The distal rib in *Lariosaurus sanxiaensis* and *Hanosaurus hupehensis* distally narrow (Figs. 4D, 4G;

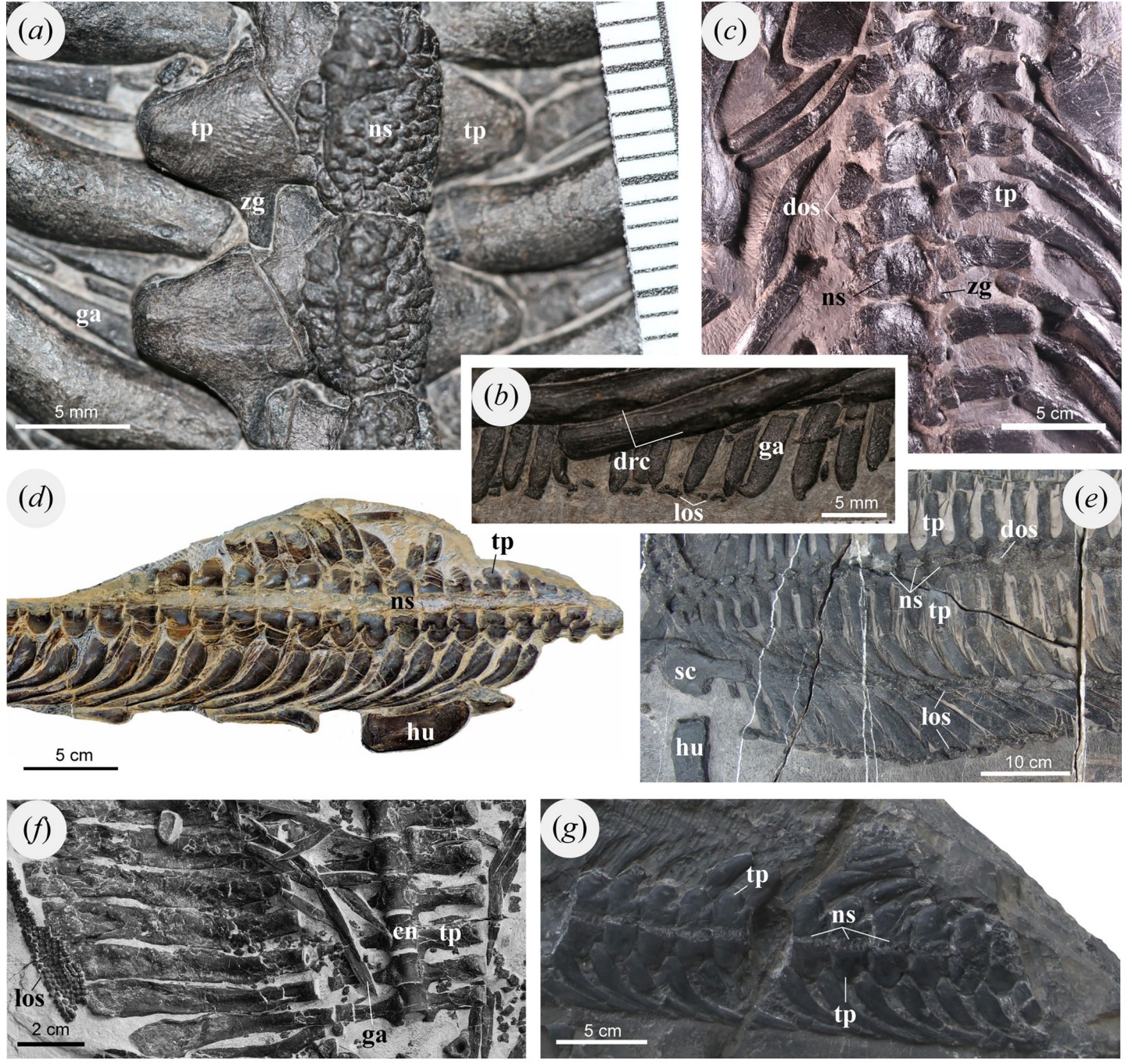

**Figure 4 Dorsal vertebrae and osteoderms of selected marine reptiles.** (A, B) *Pomolispondylus biani* gen. et sp. nov. (WGSC V1701), dorsally exposed (A) dorsal neural spine (B) crest in dorsal rib and lateral ossification. (C) *Placodus inexpectatus* (GMPKU-P-1054), dorsally exposed. (D) *Lariosaurus sanxiaensis* (HFUT YZS-16-01), dorsally exposed. (E) *Largocephalosaurus polycarpon* (WIGM SPC V 1009), dorsally exposed. (F) *Sinosaurosphargis yunguiensis* (IVPP V 17040), ventrally exposed. (G) *Hanosaurus hupehensis* (WGSC V 2010), dorsally exposed. Abbreviations: cn, centrum; dos, dorsal ossification; drc, crest in dorsal rib; ga, gastralia; hu, humerus; los, lateral ossification; ns, neural spine; sc, scapula; tp, transverse process; zg, zygapophyses.

*Rieppel, 1998*; *Chen et al., 2016*), but a small crest is seen in *Eusaurosphargis* and a larger uncinate process the placodont *Paraplacodus* (*Rieppel, 2000*; *Klein & Sichelschmidt, 2014*; *Scheyer et al., 2017*).

All dorsal ribs in the present specimen are more slender than other saurosphargiforms, so the inter-rib spacing is distinct to *Pomolispondylus biani*. The posterior two ribs present in the part specimen (WGSC V1701-1) orientated slightly anteriorly in their proximal region with a strong posterior curve strongly distally. Impressions of posterior three slender and short dorsal ribs are preserved to the right of the vertebral column with pointed distal ends (Fig. 2B). The 16$^{th}$ dorsal rib curves posteromedially like the anterior ones. The distinctly short and slender last two dorsal ribs extend anterolaterally (Fig. 2B).

## Sacral & Caudal Ribs

There are two impressions of sacral ribs in the counterpart that are almost equal in size and shape (Figs. 2B, 3C, 3D). The sacral rib is distinctly short, stout, and slightly expanded distally. There are three impressions of caudal ribs. The first caudal rib impression extends anterolaterally, with the tip close to the second sacral rib. The latter two caudal ribs are oriented laterally, with conical tips.

## Gastralia & Osteoderms

Gastralia are present associated with the vertebrae and ribs between the posterior coracoid and the anterior pubis, packed densely and forming a ventral sheet covering the abdomen (Figs. 2A, 2B, 3, 4A, 4B). Most of the gastralia are covered by the vertebral column and dorsal ribs as preserved, however, several of the most posterior gastralia are visible on the counterpart specimen (WGSC V 1701-2; Fig. 2B) and parts of other gastralia can be seen between the ribs (Figs. 2A, 3, 4A, 4B). Each gastral row consists of five elements: one medial and two lateral on each side (named first lateral and lateralmost here).

The median gastral element is relatively broad and slightly bowed with the distal tips curving posteriorly; there is no anterior process as in *Sinosaurosphargis* and *Largocephalosaurus* (*Li et al., 2011*, *2014*), which differs from *Lariosaurus sanxiaensis* and *Hanosaurus hupehensis* (*Rieppel, 1998*; *Chen et al., 2016*; *Li & Liu, 2020*). The distal tips of the medial element taper to a fine point that inserts between the immediately anterior row and the associated first lateral and lateralmost elements in the same row (Fig. 2A). Laterally the ramus extends to contact the first lateral and lateralmost elements, spreading across about one-half of the width of the trunk; the contact is close but appears to be a simple butt join. The first lateral element is the shortest and fits posterior to the medial and lateralmost elements, extending medially nearly to the trunk midline and laterally approximately ventral to the dorsal ribs. The lateralmost element is slender and straight, with an extensively tapered medial end but a blunt, rounded distal end. This configuration is similar to Saurosphargidae and Early Triassic Sauropterygia (*Rieppel, 1998*; *Li et al., 2014*).

The gastral series is very dense: medial elements contact neighboring medial elements anteroposteriorly along the trunk midline (Fig. 2B), while more distally they are inserted between the first lateral and lateralmost elements from the same gastral row and the

row immediately anterior to it (Fig. 2A). The lateralmost elements are positioned closely between rows, but do not contact each other distally as they are separated by the first lateral element. Four rows of gastralia correspond to one dorsal vertebra.

A series of small, irregularly shaped bone plates are found lateral to each gastral row that we interpret as rudimentary osteoderms (Figs. 2A, 4B). These plates form two rows along the lateral part of the gastral basket. Some of the more prominent plates are spindle shaped, and the dorsal surface develops ridge-like protrusions along the long axis (Fig. 4B). In *Sinosaurosphargis*, a bony plate covers the entire torso, possibly including the lateral ribs (Fig. 4F; *Li et al., 2011*). In *Largocephalosaurus*, the distal ribs are not connected to the osteoderms (Fig. 4E; *Li et al., 2014*), but in *Eusaurosphargis* the lateral rib contact a larger teardrop-shaped osteoderm (*Nosotti & Rieppel, 2003*; *Scheyer et al., 2017*). Osteoderms have not been reported for *Lariosaurus sanxiaensis*, *Hanosaurus hupehensis*, and *Keichousaurus yuananensis* (Figs. 4D, 4G; *Young, 1965*; *Rieppel, 1998*; *Chen et al., 2016*). *Eusaurosphargis* has larger lateral osteoderms than in WGSC V 1701 with distinct processes forming a triangle- or T-shape (*Scheyer et al., 2017*), while the osteoderms in placodonts like *Cyamodus* are also more strongly developed (*Pinna, 1992*), although notably absent in the plesiomorphic *Paraplacodus* (*Rieppel, 2000*; *Neenan, Klein & Scheyer, 2013*).

## Appendicular Skeleton

The posterior portions of the two coracoids can be seen between dorsal vertebra four and five on the part specimen (WGSC V1701-1) with a convex lateral margin (Figs. 2B, 3A, 3B), but no detail can be determined. Other appendicular elements including fragments of the possible left scapula, much of the right pubis, and fragments of the possible left ischium respectively, with their impressions, are preserved in the counterpart (WGSC V 1701-2: Fig. 2B). Only the proximal part of the left scapula is preserved in the counterpart specimen presenting the lateral process, if this is identified correctly based on its association with the humerus (Fig. 2B). The process, together with the responding impression, roughly shows the shape of the whole left scapula. The scapula impression is bow-like, with a slight angle at its mid-length where it is widest and narrows to a sharp distal terminus. This is similar in form to the scapula in dorsal view in *Largocephalosaurus qianensis* (*Li et al., 2014*), and broadly corresponds to Sauropterygia (*e.g., Placodus, Paraplacodus, Cyamodus, Dianmeisaurus*) and *Eusaurosphargis* (*Pinna, 1980*; *Rieppel, 1995, 2000*; *Shang, Li & Wu, 2017*; *Scheyer et al., 2017*).

The distal portion of the right pubis is lost, but the proximal portion is preserved in ventral view (Figs. 2A, 3C, 3D). The medial margin is strongly convex and covers around three vertebral centra. More distally the pubis narrows slightly (seen on the anterior margin). There is only a small fragment and subtle impression of the ischium, but it appears to be a short, bar-shaped element. Impressions of the humerus and femur are also present in the counterpart specimen. The humerus impression is somewhat more robust than the femur impression and has a posteriorly-directed curve (Fig. 2B), like other saurosphargiforms (*Li et al., 2011, 2014*) and Early Triassic Sauropterygia (*Rieppel, 1998*; *Chen et al., 2016*; *Li & Liu, 2020*), but little sign of a median constriction can be seen.

However, the femur is straight and expands somewhat towards the proximal end, like in most Eosauropterygia and all Saurosphargiformes (*Li et al., 2011*, *2014*; *Jiang et al., 2014*).

## PHYLOGENETIC ANALYSES

*Pomolispondylus biani* could be coded for 41/159 characters (25.8% completeness) in the matrix of *Li et al. (2014)* and for 28/213 characters (13.6% completeness) in the matrix of *Chen et al. (2014c)*. Our phylogenetic analyses recovered Saurosphargidae, as defined above, in all analyses except Analysis 2 under parsimony (Fig. S2), with *P. biani* as its immediate sister taxon (Figs. 5, 6; Figs. S1–S15). Support for Saurosphargidae is relatively high where recovered (bootstrap support ≥85%, Bremer support ≥2 under parsimony; credibility value ≥99% under Bayesian inference), but there is lower support for the clade *P. biani* + Saurosphargidae (bootstrap support 45%, Bremer support two under parsimony; credibility value ≥54%). All Bayesian inference analyses in MrBayes converged with ESS > 1,000 for all parameters (Table S3, Figs. S5, S7, S9, S11, S13, S15). Strong preference for the log-normal model was found in Analysis 2 (Bayes Factor difference = 6.62) and positive support for the log-normal model in Analysis 1 (Bayes Factor difference = 2.64), but neither the gamma nor log-normal model was preferred in Analysis 3 (Bayes Factor difference = 0.42; Table S4) (*Kass & Raftery, 1995*). For each analysis, gamma and log-normal models produced near-identical topologies and support values.

In Analysis 1, the topology recovered under parsimony largely matches *Li et al. (2014)* (Fig. 5A), however, *Helveticosaurus* and *Eusaurosphargis* are included within Sauropterygia in a more nested position alongside placodonts, Ichthyopterygia are recovered within Sauropterygia, and Saurosphargidae + *P. biani* are nested with Hupehsuchia and Thalattosauria. The topology is broadly similar under Bayesian inference (Fig. 5B; Figs. S4, S6), however, Saurosphargiformes + Hupehsuchia are positioned within Sauropterygia in a more deeply-nested position than placodonts. In Analysis 1 (Fig. 5A), Saurosphargidae have the following synapomorphies: (1) distal transverse process distinctly thickened [char. 68, state 1]; (2) pachyostosis of dorsal ribs [char. 72, state 1]; (3) deltopectoral crest absent [char. 93, state 2]; (4) median gastral rib elements two-pronged with lateral process on one side [char. 119, state 1]; while elongated and narrow transverse processes [char. 66, state 1] is also recovered for only some trees. Saurosphargiformes have the synapomorphies: (1) sacral ribs without distinct expansion of the distal head [char. 74, state 1]; (2) scapula with constriction separating the glenoid [char. 84, state 1]; and humerus curved [char 92, state 1]. Seven characters are shared between Hupehsuchia + Saurosphargiformes (see Supplemental Material) but homoplastic in these phylogenetic hypotheses, some of which are likely ecologically convergent with *Pomolispondylus biani* and other Saurosphargiformes (*e.g.*, straight femoral shaft [char. 104, state 0]; osteoderms [char. 136, state 1]; closely-associated gastral sets [char. 158, state 1]; see Supplemental Material). Constraining a monophyletic Ichthyopterygia + Hupehsuchia resulted in trees of length 611 (three steps longer) but causes Sauropterygia to collapse almost completely.

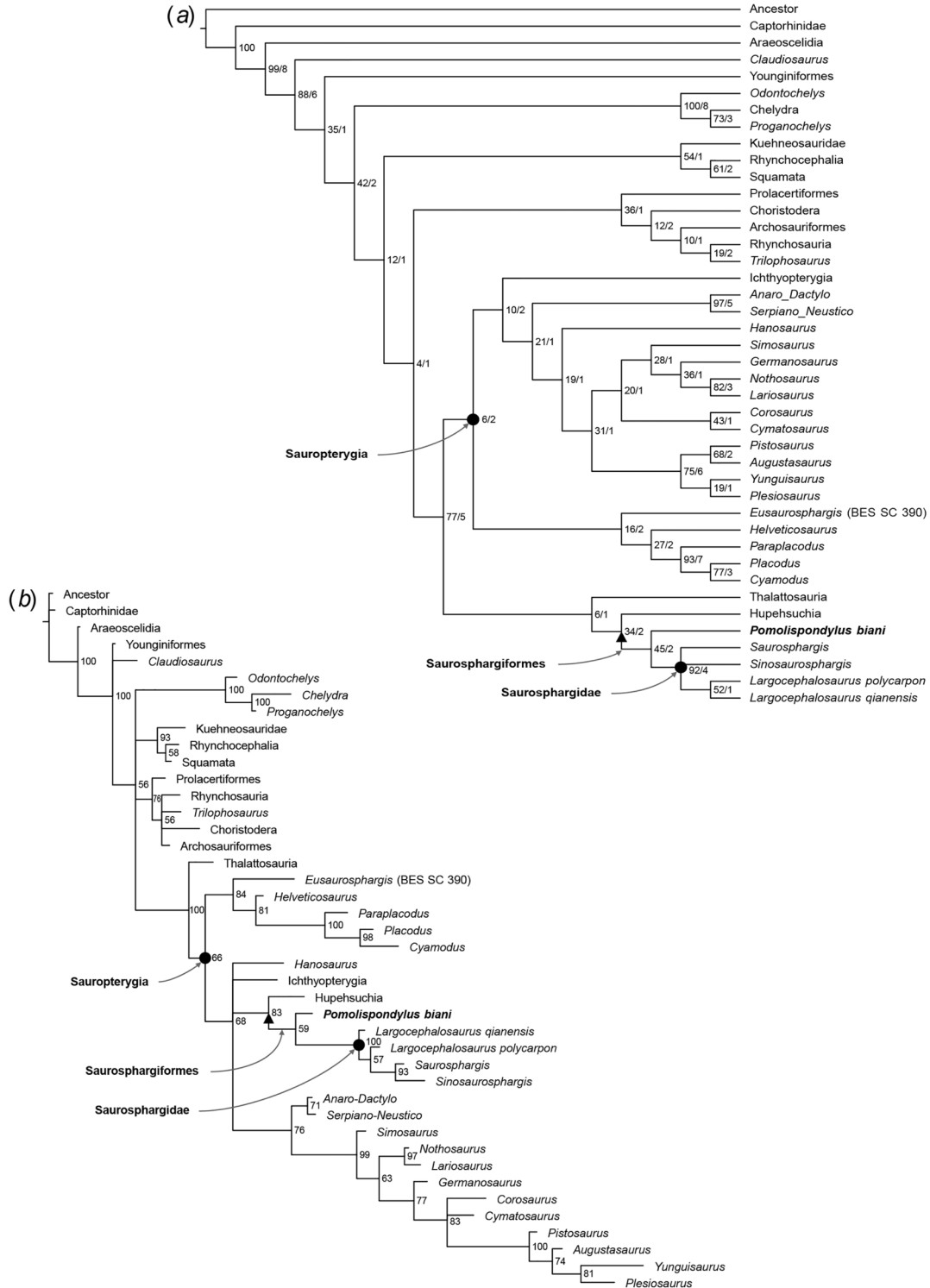

**Figure 5** Phylogenetic affinities of *Pomolispondylus biani* gen. et sp. nov. relative to other marine reptiles modified from the matrix of *Li et al.* (2014). (A) Parsimony analysis in TNT (strict consensus of 2 most parsimonious trees, length 608 steps, 77/5 bootstrap (%)/Bremer support values to right of node). (B) Bayesian inference analysis in MrBayes using log-normal-distributed rates model (50%, majority-rule consensus, 83 clade credibility values to right of node). ← denotes a maximally-inclusive clade (branch-based). ● denotes a minimally-inclusive clade (node-based).

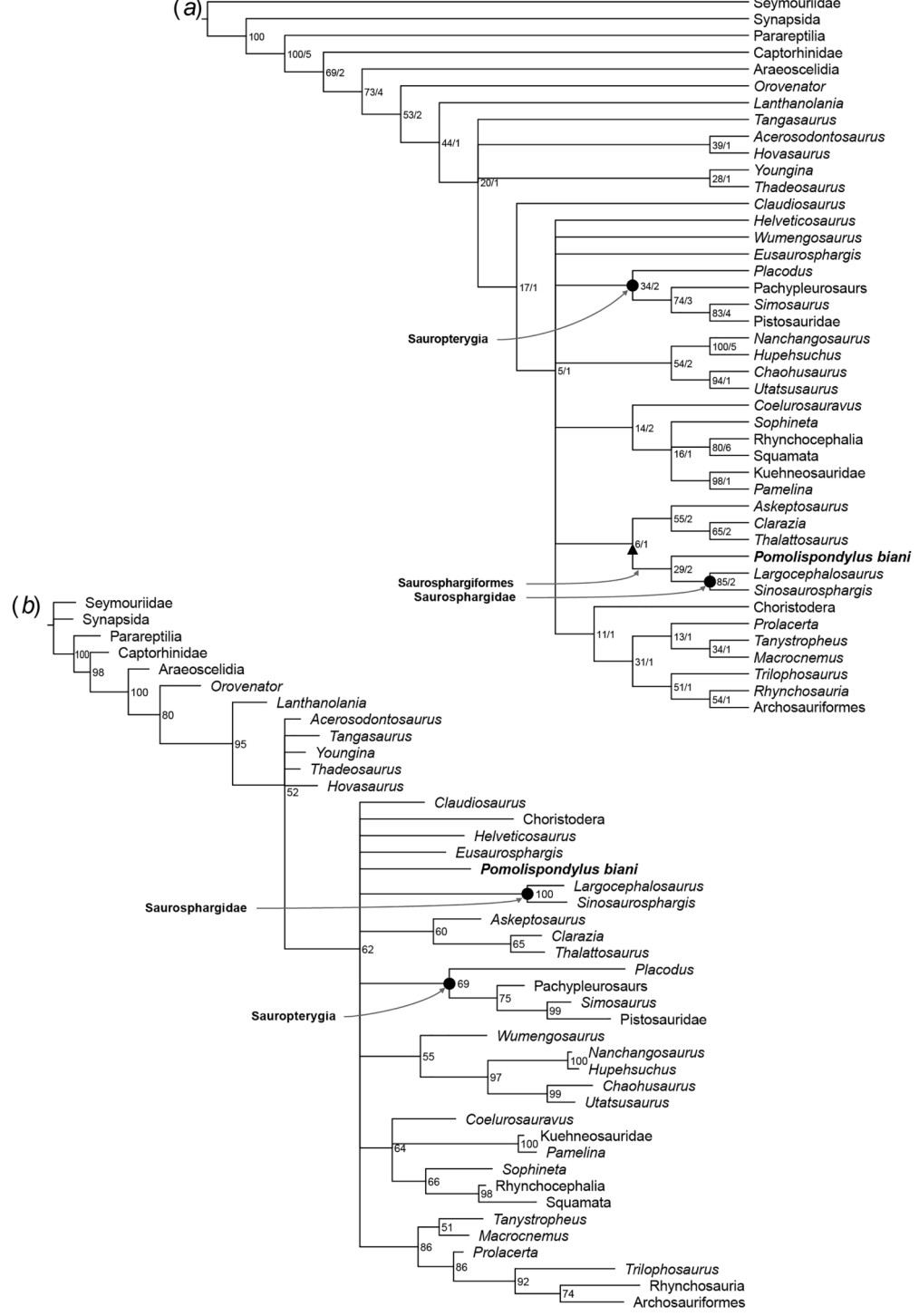

**Figure 6 Phylogenetic affinities of *Pomolispondylus biani* gen. et sp. nov. relative to other marine reptiles modified from the matrix of** *Scheyer et al. (2017)* **with marine-related characters coded as uncertainty.** (A) Parsimony analysis in TNT (strict consensus of 6 most parsimonious trees, length 814 steps, 77/5 bootstrap (%)/Bremer support values to right of node). (B) Bayesian inference analysis in MrBayes using log-normal-distributed rates model (50% majority-rule consensus, 83 clade credibility values to right of node). ← denotes a maximally-inclusive clade (branch-based). • denotes a minimally-inclusive clade (node-based).

In Analyses 2 and 3 (Fig. 6; Figs. S2, S3, S8, S10, S12, S14), the main marine reptile groups are present in a polytomous clade, with Analysis 2 additionally resolving a sister relationship between Thalattosauria and other marine reptiles (*e.g.*, Sauropterygia, Ichthyosauromorpha, Saurosphargiformes; Figs. S8, S10). Saurosphargiformes are united by: (1) the zygosphene-zygantrum articulation [char. 44], (2) curved humerus [char. 60], and (3) presence of osteoderms [char. 193], while Saurosphargidae are further united by (1) holocephalous ribs only articulating with the neural arch [char. 104], (2) pachyostosis of the dorsal ribs [char. 189], and (3) contact between the dorsal ribs forming a "rib basket" [char. 197]. Saurosphargiformes are allied with Thalattosauria by (1) a displaced mandibular articulation [char. 22], (2) slender supratemporal [char. 115], and (3) ossified atlantal ribs [char. 144].

Saurosphargiformes are not recovered as a distinct clade in Analysis 2 under maximum parsimony (Fig. S2) or Analysis 3 under Bayesian inference (Fig. 6; Figs. S12, S14). These cases correspond to a lack of resolution among the marine reptiles more generally that most likely results from the inclusion and different treatment of homoplastic characters between the two phylogenetic methods. The phylogenetic hypothesis resulting from exclusion of the marine-related characters (Fig 6; Figs. S3, S12, S14) was favored by *Chen et al. (2014c)* because these characters are frequently homoplastic, representing convergences from pressures of aquatic habitats.

## DISCUSSION

The Nanzhang-Yuan'an Fauna (NYF) is located in a restricted platform lagoonal setting, thus shallow water marine reptiles dominate (Fig. 1B; *Young & Dong, 1972*; *Chen et al., 2013*; *Cheng et al., 2015*). In recent years discovery of additional Hupehsuchia has extended the taxic and ecological diversity with features including edentulism (*e.g., Hupehsuchus, Nanchangosaurus*: *Chen et al., 2014c*, *2015*) and pachyostotic ribs for body protection (*e.g., Parahupehsuchus*: *Chen et al., 2014a*). *Pomolispondylus biani* is among the smaller species in the NYF, with an intergirdle length around 110 mm, comparable to *Keichousaurus* from South China (*Cheng et al., 2009*). Estimates by comparison to other Saurosphargiformes and Sauropterygia suggests a total presacral length around 200 mm (see Table 1 and Supplemental Material), making *P. biani* one of the smaller Early Triassic marine tetrapods (*Scheyer et al., 2014*).

*Pomolispondylus biani* shares the following traits with Sauropterygia: elongation of the neural arches, low neural spine, medially curved humerus, five-part gastralial rows. The present specimen, however, shows several differences to most Sauropterygia: a laterally-widened dorsal neural spine only found in the placodont *Placodus* (*Jiang et al., 2008*) and that is more typical of *Largocephalosaurus*, *Eusaurosphargis* (*Li et al., 2014*; *Scheyer et al., 2017*), and the stem turtle *Eorhynchochelys* (*Li et al., 2018*). Neither proximal expansion of the dorsal ribs nor the tuberosity of the dorsal neural spines occur in the holotype of *Eorhynchochelys* (*Li et al., 2018*), whereas these are present in Sauropterygia such as *Keichousaurus* (*Cheng et al., 2009*).

Despite this, there are differences between WGSC V 1701 and Saurosphargidae, including no strongly developed dorsal ossified dermal plates and shorter transverse processes of the neural spine. As this specimen is Early Triassic in age, it is older than other Saurosphargidae, known from the Middle Triassic (*Li et al., 2011*, *2014*). Although there is uncertainty in the resolution of plesiomorphic Sauropterygia in our phylogenetic analyses, a clade including Saurosphargidae and *P. biani* is recovered either within Sauropterygia (Fig. 5B) or in close relation (Figs. 5A, 6), which we name Saurosphargiformes.

The relationships between the major marine reptile groups Hupehsuchia, Thalattosauria, Sauropterygia, and Saurosphargiformes are also uncertain (Figs. 5, 6). Hupehsuchia are found as sister to Saurosphargiformes in Analysis 1 using the matrix from *Li et al. (2014)* (Fig. 5). Some of the synapomorphies of this Hupehsuchia + Saurosphargiformes clade likely represent homoplasy among ecologically-adapted characters, such as presence of osteoderms (Analysis 1, char. 136, state 1; Analyses 2 and 3, char. 193, state 1) and gastralia closely associated (Analysis 1, char. 158, state 1) both associated with the development of carapace-like structures (*Chen et al., 2014c*; *Scheyer et al., 2017*). This likely results from the focus of the phylogenetic matrix in Analysis 1 on Sauropterygia and terrestrial ancestors, so doesn't fully account for specialization of Ichthyosauriformes, particularly in the skull and limb structure, related to modification of their primary locomotory mode (*Chen et al., 2014c*; *Motani et al., 2015*; *Scheyer et al., 2017*). In the NYF, Hupehsuchia and *Hanosaurus* co-occur with *Pomolispondylus* and share morphological characters with it and Saurosphargiformes typical of the Middle Triassic: the abdominal ribs are widened laterally and are closely arranged in contact, and the pubis is rounded. The clade Thalattosauria + Hupehsuchia + Saurosphargiformes is supported in Analysis 1 (Fig. 5A) by a posteriorly displaced mandibular articulation [char. 22], slender supratemporal [char. 115], and ossified atlantal ribs [char. 144]. However, homoplasy is indicated by several reversals of characters between Thalattosauria, Hupehsuchia, and Saurosphargiformes.

The relationship between *Pomolispondylus* and Saurosphargidae allows inference of the transformation that the "carapace" underwent in Saurosphargiformes. We propose that initially cartilaginous plates developed dorsally on the neural spines and around the dorsal ribs, as in *Pomolispondylus*, then became ossified into the small bony plates found in *Largocephalosaurus* (*Cheng et al., 2012*; *Li et al., 2014*), or ossified extensively and extended across the dorsal trunk region in *Sinosaurosphargis* (*Li et al., 2011*). This is supported by recovery of osteoderms present in Saurosphargiformes as a character change (and Hupehsuchia when sister taxa: char. 136 in Analysis 1; char. 193 in Analyses 2 and 3), with subsequent modification to dense osteoderms in *Sinosaurosphargis* (Figs. 5, 6; see Supplemental Material). Presence of osteoderms in Triassic marine reptiles is relatively labile, sometimes alongside other pachyostotic morphology interpreted as adaptation to a shallow marine habitat (*Scheyer, 2007*; *Houssaye, 2009*). There remains the possibility, however, that osteoderms in placodonts and saurosphargiforms share the same

phylogenetic origin. Other Hupehsuchia from the NYF have been noted for their dorsal armor, indicating selection pressures from predation (*Chen et al., 2014a*) that may have driven the evolution of dermal ossicles in Saurosphargiformes.

This new find highlights the presence of a diverse Early Triassic fauna and the filling of different niches within the NYF, which is otherwise dominated by hupehsuchians (*Cheng et al., 2015*; *Reeves et al., 2021*). This increasingly complex ecosystem marks the beginnings of recovery following the EPME and establishing the reptile-dominated ecosystems found through the Mesozoic. The NYF also supports a different complement of taxa than other Early Triassic localities (*e.g.*, Majiashan, Anhui, China (*Jiang et al., 2014*; *Zhou et al., 2017*); Spitsbergen, Svalbard (*Maxwell & Kear, 2013*; *Ekeheien et al., 2018*)), particularly in the presence of diverse Hupehsuchia and small reptile taxa (*i.e., Keichousaurus* and *Pomolispondylus*). There is currently no evidence for large macropredators above 2 m in the NYF, which have been identified in other Early Triassic faunas (*Scheyer et al., 2014*). Hupehsuchia are the largest representatives of the NYF, but have an unusual slender "duck-billed" skull (*Qiao, Iijima & Liu, 2020*), while *Lariosaurus sanxiaensis* and *Hanosaurus* are the largest piscivores (*Cheng et al., 2015*; *Li & Liu, 2020*). As the NYF represents a restricted lagoon, this may skew the feeding modes present. The lagoonal paleoenvironment may have caused the lack of fishes and rarity of invertebrate remains if restricted flow led to occasional or more continuous harsh conditions. A similar pattern has been found for hypersaline conditions in the Permian (*Piñeiro et al., 2012*). It is possible that taxa in the NYF were particularly adapted to these conditions, hence their restricted geographic range. The lack of fish and macroinvertebrates in the NYF precludes establishing the full food web, however, the diversity of sizes and feeding modes supports limited but productive ecosystem.

This reptile diversity has been argued to indicate the earlier recovery of marine vertebrate ecosystems than previously thought (*Jiang et al., 2014*), however, the relative abbreviation of food chains found in these Early Triassic ecosystems instead supports changing constructions of ecosystems between the Early and Middle Triassic and a more delayed recovery (*Benton et al., 2013*; *Cheng et al., 2015*; *Song, Wignall & Dunhill, 2018*; *Li & Liu, 2020*). The size and morphology of *P. biani* follows from infilling of ecological niches while there is spare carrying capacity prior to the building up of the ecosystem (*Song, Wignall & Dunhill, 2018*; *Reeves et al., 2021*), however, the geographical restriction and temporal brevity in which the taxa present in the NYF are found otherwise may indicate the brief flourishing of a transitional fauna before later turnover. Finds of the NYF outside this limited time and space region would discount this second hypothesis.

## ACKNOWLEDGEMENTS

We thank Michael Benton (University of Bristol), Xiao-chun Wu (Canadian Museum of Nature), and Li Tian (China University of Geosciences, Wuhan) for discussion on this specimen. We also thank Dongyi Niu for fossil collection and preparation. We also thank Valentin Buffa, Donald Brinkmann, and two anonymous reviewers for their comments.

## Funding

This work was supported by the China National Natural Science Foundation grant (No. 41972014), the China Geological Survey (Nos. DD20190811 and DD20221634), the NERC BETR grant (NE/P013724/1) and the ERC grant (788203). The funders had no role in study design, data collection and analysis, decision to publish, or preparation of the manuscript.

## Grant Disclosures

The following grant information was disclosed by the authors:
National Natural Science Foundation, China: 41972014.
Geological Survey, China: DD20190811 and DD20190315.
NERC BETR: NE/P013724/1.
ERC: 788203.

## Competing Interests

The authors declare that they have no competing interests.

## Author Contributions

- Long Cheng conceived and designed the experiments, analyzed the data, prepared figures and/or tables, and approved the final draft.
- Benjamin C. Moon performed the experiments, analyzed the data, prepared figures and/or tables, and approved the final draft.
- Chunbo Yan analyzed the data, authored or reviewed drafts of the article, and approved the final draft.
- Ryosuke Motani analyzed the data, authored or reviewed drafts of the article, and approved the final draft.
- Dayong Jiang analyzed the data, authored or reviewed drafts of the article, and approved the final draft.
- Zhihui An analyzed the data, authored or reviewed drafts of the article, and approved the final draft.
- Zichen Fang analyzed the data, authored or reviewed drafts of the article, and approved the final draft.

## Data Availability

The raw data is available in the Supplemental Files.

## New Species Registration

The following information was supplied regarding the registration of a newly described species:
Publication LSID: urn:lsid:zoobank.org:pub:2EF3BA4D-B401-4B86-88C9-9320466A1ACD

![PeerJ]

*Pomolispondylus biani* gen. et . sp. nov. urn:lsid:zoobank.org:act:42A625CB-EE27-4432-8606-7DD0ADC760D4.

## Supplemental Information

Supplemental information for this article can be found online at http://dx.doi.org/10.7717/peerj.13569#supplemental-information.

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
