# Peer review of "The oldest record of Saurosphargiformes (Diapsida) from South China could fill an ecological gap in the Early Triassic biotic recovery"

_PeerJ, doi:10.7717/peerj.13569_

## Round 0.1 · original submission · Major Revisions

Dear authors,

We have now four review reports for your manuscript on “The oldest record of Saurosphargiformes (Diapsida) from South China fills an ecological gap in the Early Triassic biotic recovery” and all of them consider your manuscript interesting and within the scope of PeerJ.

However, several changes will need to be made to the manuscript, including revision of data and procedures for the phylogenetic analysis and the nomenclatural acts, as well as the consideration of some changes that should be performed to the figures and figure captions plus the expansion of some sections of the manuscript following strictly the recommendations and commentaries from the reviewers.

Particularly important is that you include anatomical comparisons of the new taxon with respect to the previously discovered diapsid materials from the same paleontological site and that you consider providing a geological setting section. Otherwise, you can send a rebuttal letter explaining your point of view and the arguments by which you disagree with the requested modifications. However, I strongly recommend that you consider accepting every criticism from the reviewers because all of them are constructive and would enhance the relevance of your work in the field of early diapsid evolution.

Once you complete the requested changes, please resubmit your revised pdf for a new round of considerations and decisions.

With my best regards,
Graciela Piñeiro

·

Excellent Review

This review has been rated excellent by staff (in the top 15% of reviews)
EDITOR COMMENT
I agree with the decision of selecting this review report as an excellent input for improving the manuscript. It is perfectly clear, ordered and complete, denoting the expertise of the reviewer in particular about the cladistic methodology and phylogenetic relationships of the involved diapsid groups but also on the identification of the general guidelines that are needed to demonstrate that authors are engaged with the topic being discussed.

Basic reporting

This is a nice descriptive work of an interesting new taxon. However, I have several issues with the phylogeny and discussion presented in this manuscript.

Experimental design

See comments below.

Validity of the findings

No comment.

Additional comments

The comments listed here follow the order of the manuscript with the main sections indicated.

Materials
- Some information written in the “locality and horizon” subsection of the Systematic Paleontology could be added here.

Phylogenetic analysis
- I did not expect such drastic changes in topology between the PAUP* and TNT analyses as figured in the Supplementary Material (SM). Indeed, I re-ran the PAUP* analyses using the methods described in the SM and found the same topologies as the TNT ones, with the number of MTPs and MTP lengths recovered in the PAUP* figure captions in the SM. Could the trees in figs. S5-S8 be bootstrap consensus trees instead of the strict consensus trees resulting from the parsimony analyses? I admit I am unsure.
- I suggest re-running the PAUP* analyses to clear up this inconsistency. This is particularly important as the PAUP* analyses are considered as the “primary analyses” in the SM, yet are barely mentioned and not figured in the main text. Furthermore, they seem to be problematic for the discussion due to their apparent irresolution.
- Provided that the PAUP* and TNT analyses are indeed similar, I would say that the four analyses conducted by the authors are unnecessary. Analysis 1 is but a subset of analysis 3, and seems more pertinent in studying the systematic position of Hanosaurus and Hupehsuchia, which is out of the scope of this paper. Similarly, I see little point in analysis 2. Analysis 4 has some merit in excluding Ichthyopterygia as the present matrix does not consider the specializations of this group, but I do not really understand why Hanosaurus is excluded as well. Hence, I would be inclined to keep only analysis 3 (42 OTUs), and possibly analysis 4 (but keeping Hanosaurus) to discuss the surprising recovering of Ichthyopterygia inside Sauropterygia. If the authors wish to retain these different analyses as they are, then an explanation of why each taxon is included/excluded is required.
- Optional, but I urge the authors to consider ordering multistate characters when the character states form morphoclines. Simulations and theoretical considerations indeed indicate that the results are better than when all characters are unordered, even if minor mistakes in ordering are made (Rineau et al., 2015, 2018). This might help in solving the ‘problematic’ position of Ichthyopterygia inside Sauropterygia, although I understand that this is out of the scope of this paper.

Rineau, V., A. Grand, R. Zaragüeta i Bagils, and M. Laurin. 2015. Experimental systematics: sensitivity of cladistic methods to polarization and character ordering schemes. Contributions to Zoology 84:129–148.
Rineau, V., R. Zaragüeta i Bagils, and M. Laurin. 2018. Impact of errors on cladistic inference: simulation-based comparison between parsimony and three-taxon analysis. Contributions to Zoology 87:25–40.

Systematic Palaeontology
- A revised phylogenetic definition of Diapsida has been published recently: Gauthier, J. A., and K. de Queiroz. 2020. Diapsida; pp. 1033–1040 in K. de Queiroz, P. Cantino, and J. A. Gauthier (eds.), Phylonyms: A Companion to the PhyloCode. CRC Press, Boca Raton.
- Taxonomic authorities should be given for the taxa used in the newly proposed phylogenetic definitions.
- The diagnosis of Tuberospina should only include diagnostic characters among saurosphargiformes based on the comparative description and/or the phylogenetic analysis, which does not seem to be the case here.

Description
- I strongly suggest adding a short paragraph assessing the maturity of the specimen. This is particularly important as its small size supports the idea that it filled a new ecological niche.
- There seems to be a confusion in the description of the pectoral and pelvic girdles. Elements of the clavicle and ilium are described but figured as the scapula and ischium respectively. Please correct accordingly.

Phylogenetic analyses
- See comment in methods above, I believe a portion of this section will have to be corrected.

Discussion
- lines 280-287: this section needs to be revised provided the PAUP* and TNT analyses are indeed similar. As suggested above, only analysis 3 (and possibly analysis 4) could be kept and figured in the main text.
- lines 288-289, and 305-327: Tuberospina comes from ca. 10m lower in the sequence than other marine reptiles (Fig. 1). Can both horizons really be considered coeval? Is this correlation based on stratigraphic studies? I am not opposed to the idea of the taxa being coeval (though I would say the specimens are not), but this needs to be argued before the NYF can be convincingly discussed as a whole.
- lines 294-304: Provided the PAUP* and TNT analyses are indeed similar, then Hupehsuchia is consistently recovered as sister-group to Saurosphargiformes in all analyses. I feel a comparison with hupehsuchians is necessary to complete the proposed scenario of evolution of dermal armor in this lineage.

Some minor comments:
- lines 143-146: I suggest adding a caption for this cervical rib on Fig. 2.
- line 159: please caption the zygosphene-zygantrum on Fig. 3.
- lines 166-167: to what do these ratios correspond?
- line 180: I am inclined to agree that the dorsal ribs seem single-headed, but was unable to confirm it on the figures. Please explain how you reached this conclusion.
- line 187: refer to Fig. 2D instead of Fig. 3A, B.
- lines 198-200: “three impressions of caudal ribs”, but only one is captioned on Figs. 2, 3. Please caption the other two rib impression, if present.
- line 222: refer to Fig. 2D instead of Fig. 2A
- line 241: “humerus somewhat longer and larger than the femur”, could this width difference be an artefact of the partial impression of one or both of these bones in the matrix?
- line 259: optional, but I would repeat what NYF means here for an easier reading.
- line 261: “toothless forms, pachyostotic ribs, and other features” this list is confusing, please rephrase.
- line 267: “development of the neural arches”. How exactly? This is ambiguously worded.
- lines 268-271: I am not sure I understand the logical link after the “;” line 270.
- lines 290-291: “sister relationship Tuberospina + Hupehsuchia” I don’t understand this. Do you mean they are recovered as successive sister-groups to saurosphargids? Please rephrase.

Figure captions
- Tuberospina biani should be italicized in all captions.
- Fig. 1: there is no reference to the colored (?paleogographical) map in the top right corner.
- Fig. 2: I suggest removing “part specimen (WGSC V1707-1-1) from the bold portion for consistency with the counterpart caption.
- Fig. 2: there is no caption for subfigs. 2C and 2D. Some of the references in the main text may have to be corrected to refer to those subfigs.
- Fig. 3 (line 485): remove “and”.
- Fig. 3: change “tp” for “tr” as written on the figure, or vice-versa.
- Fig. 4: Please indicate that only a selected portion of the tree is shown on the figure, and state which type of consensus tree is shown.

·

Basic reporting

The only edits I would suggest are:

on line 41: change "contrasts terrestrial' to 'contrast with terrestrial'.

in the figure legends, the name "Tuerospina biani" should be in italics.

Experimental design

no comment

Validity of the findings

no comment

Additional comments

Early Triassic marine reptiles are extremely rare. The specimen described in this paper provides a significant addition to our knowledge of marine reptiles of this time and provides additional information on a poorly known lineage of Triassic marine reptiles. Writing is clear and concise. The importance of the specimen within the context of the initial diversification of marine reptiles of the Triassic is clearly explained. Apart for the minor edits noted above, I saw no need for revisions to the manuscript and am recommending publication without further modification.

Reviewer 3 ·

Excellent Review

This review has been rated excellent by staff (in the top 15% of reviews)
EDITOR COMMENT
Indeed, the report from this reviewer was excellent, not very often editors count with such clear and very helpful revisions for guiding authors to improve their work. It encompassed almost all the manuscript sections, focusing mainly on the taxonomic results provided by the authors, an aspect which is fundamental to perform a correct identification of the described material and to perform a correct phylogenetic study.

Basic reporting

The manuscript is well written, with clear, unambiguous and professional English used throughout the manuscript, but please see some typo here and there as shown below in the additional comments. The literature is appropriately referenced. Sufficient field background is provided. The article structure is professional. However, the figures should definitely be improved. Please see additional comments below for details. The raw data have been provided. The manuscript is self-contained with relevant results to hypotheses.

Experimental design

The research is original and within Aims and Scope of the journal. The research question is well defined, relevant & meaningful, and it is clearly stated how the research fills the identified knowledge gap. The investigation should be improved as stated below in additional comments. The research was conducted in conformity with the prevailing ethical standards in the field. Methods were described with sufficient information.

Validity of the findings

The currently provided evidence is not enough to support some of the major conclusions of the research. Please see below for details.

Additional comments

1. Line 13: numbers don't correspond with those in author list.

2. Line 16: number doesn't correspond with those in author list.

3. Line 23: the genus name is pre-occupied by a shark, see below.

4. Line 24: it is meaningless to create Saurosphargiformes here.

5. Line 70-71: first, why only Chaohusaurus zhangjiawanensis is emphasized here; second, publications on sauropterygians from the fauna should also be cited here.

6. Line 73: if it shares critical synapomorphies with Saurosphargidae, shouldn't it be a saurosphargid? What are the synapomorphies of Saurosphargidae? This isn't given anywhere in the text.

7. Line 79-80: the fossil locality name reported in the text should be consistent with the locality name in Fig. 1 and its caption.

8. Methods: this part does not look like a description of methods used in the research. It should be more appropriate to move this part to a note somewhere in the end of the manuscript.

9. Line 97: so why wasn't the phylogenetic analysis performed on the basis of this matrix (Chen et al. 2014c, and its updated version in Motani et al. 2015 and Jiang et al. 2016), but on the matrix of Li et al. (2014) instead, even though it doesn't address these potential convergences? Compared with Li et al. (2014)’s data matrix, which is updated on Li et al. (2011) but originally derived from Nosotti and Rieppel (2003), the data matrix of (Chen et al. 2014c, Motani et al. 2015 and Jiang et al. 2016) is much expanded and more appropriate to analyze the phylogenetic relationships of Saurosphargidae. In addition, many other updated phylogenetic analyses also include Saurosphargidae (e.g., Li et al., 2018; Schoch and Sues, 2018). The authors need to justify their choice of Li et al. (2014)’s data matrix in the manuscript.

10. Line 111: please add diagnosis of Saurosphargidae.

11. Line 119: the genus name is pre-occupied by Tuberospina LEBEDEV 1995, a Devonian shark from Russia.

12. Line 126: “WGSV V1701-1” to “WGSC V1701-1”.

13. Line 124-126: it would be interesting to know what happened to the skull and the tail of the specimen?

14. Line 131: the new specimen is said to be collected from Member II of the Jialingjiang Formation in the manuscript, but in a series of papers by the same group (Chen et al., 2014a, b,c 2015, Cheng et al. 2019), the marine reptile horizon is in the third member of Jialingjiang Formation. Is this new specimen from a different horizon? The authors need to provide some explanations to clarify this.

15. Description: the entire section needs more comparisons with pachypleurosaurs (and nothosaurs), to which the specimen is at least superficially similar. I think the authors should also provide and label an additional figure, comparing the dorsal regions (in dorsal view) between the new specimen, Largocephalosaurus, a pachypleurosaur (for example Hanosaurus from the same region), a nothosaur (for example Lariosaurus sanxiaensis from the same region) and a placodont to make the differences and similarities in anatomy clearer to the reader.

16. Line 148-150: there are Hanosaurus, Keichousaurus and Lariosaurus from the same region. Why there is no comparison with these eosauropterygians? They look very similar to the new specimen reported here.

17. Dorsal Vertebrae: dorsal Vertebrae of the new specimen look very similar to Lariosaurus sanxiaensis from the same region. Actually the dorsal Vertebrae of the new specimen is a very general morphology of pachypleurosaurs and some of the nothosaurs, both of which are present in the same fauna. The rugose dorsal surface of the neural spine is also similar to Augustasaurus, if not to all eosauropterygians.

18. Line 177: ‘vertebrate’ to ‘vertebral’.

19. Line 185-187: I do not really understand the so-called crest. The authors should figure this structure.

20. Line 187: ‘Fig. 3A, B’ to ‘Fig. 3a, b’.

21: Gastralia & Osteoderms: please provide at least some comparison with gastralia of other saurosphargids and sauropterygians.

22: Line 206: ‘laterally’ to ‘lateral’.

23: Line 209: if there is anterior process cannot be confirmed from the photos. Please provide a close-up photograph.

24: Line 222: please provide a CLOSE-UP photo of these rudimentary osteoderms- impossible to confirm what they are from photos currently provided.

25: Line 229: ‘vertebral’ to ‘vertebra’.

26: Line 232: but the left clavicle is labeled as a scapular in Fig. 2B? The authors should provide a reconstructed clavicle based on the composite images.

27: why the ilium is not figured?

28: PHYLOGENETIC ANALYSES: an additional analysis based on both scenarios of Chen et al. (2014, updated in Motani et al. 2015 and Jiang et al. 2016) (all morphological characters + aquatic adaptations treated as ambiguous) should be added to the manuscript, perhaps with the addition of several pachypleurosaurs. The leading author (Rysouke Motani) of those papers can easily re-run the analysis.

29: Line 247: what character states support the node of Saurosphargidae?

30: Line 254: Fig. 4C is different from what described here.

31: Line 267-273: the authors mention many similarities to pachypleurosaurs in the paragraph. So why the new specimen is not a pachypleurosaur? It would be good to include a panel with photo showing the dorsal region of Placodus, pachypleurosaurs, nothosaurs, saurosphargids, and the primitive turtles for comparison.

32: Line 277-279: none of the sub-figures of Fig. 4 shows a sister-relationship between the so-called Saurosphargiformes and the Sauropterygia. Instead, all figures show that Hupehsuchia (when it is shown in the tree) comprises the immediate sister group of Saurosphargiformes (Fig. 4b, c).

33: Line 284-286: so again, why didn’t you test this using two versions of the data matrix of Chen et al 2014) and Motani et al. (2015)?

34: Line 288-290: the authors provide another line of evidence that the new specimen could be a pachypleurosaur.

35: Line 290-293: again, test this by performing a phylogenetic analysis.

36: Line 311: if Keichousaurus is present in the locality you definitely need to include comparisons with Keichousaurus and include it into a phylogenetic analysis.

37: Line 312-315: what’s the definition of larger macropredators? What’s the largest size of Lariosaurus sanxiaensis from the fauna? Isn’t Lariosaurus sanxiaensis larger than Hanosaurus?

38: Fig. 1: ‘WGSC C1701’ to ‘WGSC V1701)’.
For panel A, please note that the name of the fossil locality in the figure is different from the text. For panel B, I understand that the new specimen is from Bed 19, but what for those other marine reptiles? Are all of them from Bed 27? Previous publications from the same group indicate that marine reptiles are present all through the section.
For panel C, this is a nice paleogeographic reconstruction. I presume that this is the result from the authors’ field mapping activities since no reference is cited. But where is the evidence for this paleogeographic reconstruction? And also the fossil locality is located in the open platform judged from the color, but the authors repeatedly mentioned in the manuscript that the fauna is from a restricted lagoon. So which is correct?

39. Fig. 2: what about panels C and D? they're not explained in the caption. The authors need to provide a close-up photo showing the lateral ossifications. The current photo is far from clear to judge if these are osteoderms or not.

Reviewer 4 ·

Basic reporting

The manuscript is well structured, the language is professional and understandable.

The cited literature is appropriate in terms of content and form. I would suggest a small addition for the geological background (see Additional comments).

The figures are correct, in some cases, small corrections are needed for the captions (see Additional comments).

The data files used for the analysis are available and interpretable.

The anatomical description and phylogenetic analysis are professionally demanding, however, I made a comment on the analysis.

Experimental design

I found the 128. character of Tuberospina problematic in the matrix (durophagous dentition etc.). Even though the skull is unknown, the character is coded as “absent” instead of “?”, which I think it should be (as the other cranial characters).

Probably it is just a coincidence error with minimal impact, but I would like to ask the authors to reconsider this and in case there is no other explanation I would suggest checking and repeating the analysis with the corrected matrix. Furthermore, it would be advisable to double-check the matrix.

Validity of the findings

no comment

Additional comments

line 80 - Because of the clarity, I would also mention the name of the quarry here.

line 81 - Although the relevant references are cited in the text, and Figure 1. is clear, I think it would be useful to give a longer (but brief) geological description of the formation and sequence where the specimen was found (especially about the stratigraphic background).

line 471 - (Fig.1 caption) The “C” is probably a misspelling, please change it to WGSC V1701. I would suggest mentioning that the Early Triassic paleogeographic map of the surrounding region is also visible here.

line 476 - (Fig.2 caption) Please provide a caption for the C and D parts of the figure as well. In case the contents of C are visible on the A or B picture, I suggest marking its location.

line 489 - (Fig.3 caption) On the figure it is tr instead of tp., please change it to be consistent.

---

## Round 0.2 · Minor Revisions

Dear authors,

I have sent your manuscript for a second review and we have now the new reports, which although coincident in that the manuscript was significantly improved, some problems remain. Indeed, the phylogenetic study and the figure captions seem to be the sections which need the most careful attention. I think that if you follow the recommendations from the Reviewers, you can be able to resolve it.
But the issues with the figures and the figure captions are relevant to me, because if you do not provide enough clear figure captions and labeling of the characters that show the different taxa that you are comparing, I cannot make a comparative evaluation of what you are providing and what the reviewers (which are specialists in the area) require.

Therefore, I suggest that you revise again the phylogenetic section, regarding each of the reviewer’s recommendations, making all of the reviewer’s requests (including the reconsideration of the matrix to choice and the addition of Keichousaurus yuananensis into the phylogenetic analysis).

All the questions and concerns included in the review reports have to be properly answered in a rebuttal letter (including the geological setting requests for clarification), and if corresponds, the resulted answers should be part of the manuscript text.

Moreover, I expect to see a new version of your manuscript with the figures well explained and referred in the text and particularly I request that you label each character that you consider diagnostic and that you include an explanation for them in a well clear figure caption. For instance, the presence of osteoderms in the new proposed taxon must be guaranteed from the figures and a label for their position on the figure is required. The same procedure is required by the other diagnostic characters on which you are based the presence of a new taxon.

I expect to see the revised version of your manuscript soon, addressing all the concerns and recommendations from the reviewers which are clearly explained in the new review reports and remarked in my decision letter.

Kind regards,
Graciela Piñeiro

·

Basic reporting

The revisions have been adequately addressed for the most part. I have only minor comments on the well-written description of this very interesting specimen. However, the authors have significantly revised the phylogenetical analyses, on which I have further issues.

Experimental design

See comments below.

Validity of the findings

No comment.

Additional comments

Methods
- To be honest, I do not understand the choice of Li et al.’s (2018) matrix for analysis 2 over Jiang et al.’s (2016) as suggested by Reviewer 3. The matrix of Li et al. (2018) seems very ill-suited for the question at hand in my opinion as it lacks critical taxa (e.g. hupehsuchians, thalattosaurs, ichtyopterygians), and I do not believe it allows for a reliable discussion of the relationships of saurosphargids to other marine diapsids (see comments on the Discussion). I thus urge the authors to further argue their choice of matrices, and/or to reconsider their choice of Li et al.’s (2018) matrix.

Systematic Palaeontology
- I do not understand how the characters provided in the diagnoses of Saurosphargiformes, Saurosphargidae and Strumospina were obtained. The authors provide a list of synapomorphies for Saurosphargidae lines 338-346, yet none of the characters recovered from the phylogenetic analyses are reported in the diagnosis, while none of the ‘diagnostic’ characters provided in the systematic paleontology are recovered as synapomorphies.
- The authors have modified the diagnosis of Strumospina but the issues I had still remain: some characters do not seem diagnostic, especially after reading the description (e.g. number of vertebrae, curvature of humerus…), or are not described in the text (e.g. vertebral centra with parallel edges, oval facets on ribs…).
- Ontogenetic assessment: Thank you for adding this important paragraph, however, I believe it should be either part of the description of an independent section, not part of the Systematic Paleontology.

Phylogenetic Analyses & Discussion
- The phylogenetic results and discussion are hard to follow as the authors seem to go back and forth between analyses 1 and 2 with analyses 3 and 4 being marginal despite their slightly different results. I suggest being more explicit on the results from all analyses to clarify the text.
- A quick description of the relevant branches and nodes is needed. For instance, the inclusion or exclusion of saurosphargids from Sauropterygia seems highly relevant to me. Indeed, while Fig. 5B and 6A show saurosphargids inside Sauropterygia, Figs. 5A and 6B do not. Thus, I disagree that Saurosphargiformes are included in Sauropterygia based only on the results provided here, contrary to lines 370-373.
- The authors only present the synapomorphies for Saurosphargidae under analysis 1. What of the other analyses? And what of Saurosphargiformes and Strumospina? As mentioned above, these characters, if supported by all analyses, should be included in the diagnoses.
- Various characters are mentioned throughout the discussion. Are those synapomorphies recovered from the phylogenetic analyses? From all analyses? This needs to be more explicit.

- lines 390-400: What exactly is the plesiomorphic state for the Saurosphariformes regarding the evolution of the “carapace”? Without this information, there are many scenarios that are equally parsimonious, including a reduction of the “carapace” in Strumospina. This is particularly important as this morphology seems highly convergent in Traissic taxa. I made a similar comment on the first draft, which went unanswered. Please address this issue, otherwise I see little point in this discussion.

- I paste here another remark I made on the first draft which has not been responded to: “lines [383-384], and [401-423]: [Strumospina] comes from ca. 10m lower in the sequence than other marine reptiles (Fig. 1). Can both horizons really be considered coeval? Is this correlation based on stratigraphic studies? I am not opposed to the idea of the taxa being coeval (though I would say the specimens are not), but this needs to be argued before the NYF can be convincingly discussed as a whole.”


Some minor comments:
- line 99: remove one “)”
- line 145: please provide translation for Latin words
- line 331 this should refer to Fig. 5. There is thus no reference to Figure 4, please add those
- line 378: change “gastral” for “gastralia”


Figure captions
- The pdf file has several issues with the captions, but those are absent in the word file. A quirk of the proofs?
- Fig. 2D is no longer present but still captioned.
- Fig. 3: the zygosphene-zygantrum arrow points to the zygapophyses, please correct this.
- Fig. 4: explanations for ‘hu’ and ‘sc’ are missing.
- Table 1 is still present as a table caption in the ms although it is a supplementary table

Reviewer 3 ·

Basic reporting

See the comments below.

Experimental design

See the comments below.

Validity of the findings

See the comments below.

Additional comments

The authors have considered most of the comments from the four reviewers and revised the manuscript accordingly. I feel that the revised manuscript is significantly improved. However, there are still several major issues that need to be addressed before the recommendation for publication.

My major concern is the validity of the new taxon when compared with Keichousaurus yuananensis from the same fauna. Based on the description and comparison present in the manuscript, I do not see any difference between Keichousaurus yuananensis and the proposed new taxon. First, the size of the new specimen is comparable to the only known type of Keichousaurus yuananensis (150 mm vs 145 mm in the dorsal region). Second, the number of dorsal vertebrae in the new specimen is similar to (or even same with) Keichousaurus yuananensis (~18 vs ~19-20). Third, the humerus of the new specimen shares a characterized shape with Keichousaurus yuananensis holotype. In the manuscript, the only noted difference between the two specimens is the osteoderm. Osteoderms are present in the new specimen based on the updated photos, but its absence in Keichousaurus yuananensis holotype can’t be confirmed since only bone impression is known in the holotype of Keichousaurus yuananensis. For example, osteoderms cannot be confidently identified from the part specimen (WGSC V1701-1) where only impression is preserved. One way for sorting this out would be the addition of Keichousaurus yuananensis into the phylogenetic analysis, coding osteoderm characters as question marks. If the phylogenetic analysis can not separate the two OTUs, it is very likely they are the same species.

Other minor issues:
1. Etymology: What does ‘struma’ mean? I only found definitions indicating that ‘struma’ is either a swelling of some soft tissues or tumors of soft tissues – unless the authors provide a specific definition of ‘struma’ that they are referring to, using such a name will be very inappropriate.
2. Diagnosis of the new taxon: The centrum has lateral surfaces, not lateral edges.
3. The following description and comparison of the neural spines is difficult to be understood: Horizontally near the neural arch, particularly anteroposteriorly (Fig. 3a, b) , so that the neighboring two neural spines contact each other on their anterior and posterior margins. However, the neural spine differs from nothosaurs, Hanosaurus hupehensis, and pachypleurosaurs by quickly expanding as each becomes laterally broader towards their posterior too (Fig. 2a, b; Young, 1965; Rieppel, 1998).
4. The caption for Fig. 2c is still missing from the revised manuscript!!!!!!
5. The caption of Fig. 4 (phylogenetic affinities) does not correspond to the figure itself (photos of the specimen). Same applies to Figs. 5 and 6 where there must be something wrong for the figure caption.

In addition, below are some of the reviewers’ comments to which the authors did not provide any response in the response letter, nor updated in the revised manuscript (or maybe I miss). I feel that these are reasonable comments and should be addressed rather than simply ignored.
Reviewer 1:
- lines 288-289, and 305-327: Tuberospina comes from ca. 10 m lower in the sequence than other marine reptiles (Fig. 1). Can both horizons really be considered coeval? Is this correlation based on stratigraphic studies? I am not opposed to the idea of the taxa being coeval (though I would say the specimens are not), but this needs to be argued before the NYF can be convincingly discussed as a whole.
Reviewer 3:
For Fig. 1C, this is a nice paleogeographic reconstruction. I presume that this is the result from the authors’ field mapping activities since no reference is cited. But where is the evidence for this paleogeographic reconstruction? And also the fossil locality is located in the open platform judged from the color, but the authors repeatedly mentioned in the manuscript that the fauna is from a restricted lagoon. So which is correct?
Reviewer 4:
line 81 - Although the relevant references are cited in the text, and Figure 1. is clear, I think it would be useful to give a longer (but brief) geological description of the formation and sequence where the specimen was found (especially about the stratigraphic background).

---

## Round 0.3 · Minor Revisions

Dear authors,

I have carefully reviewed the new versión of your manuscript and I think that it was strongly improved by having considered the comments that you received from the reviewers. Figures and captions were also very improved by adding labeling pointing to the diagnostic structures and characters. Phylogenetic studies look more consistent now, although as you should know, the results would change in the future by the addition of more taxa or characters or the application of different methodologies and software. I just recommend some additional changes to the text that you can see in the attached annotated pdf.

Please, verify that all the references are in the Reference section and in the text; I will give a revision of that in the next submitted new version of the manuscript.

With my best regards,
Graciela Piñeiro

---

## Round 0.4 · accepted · Accept

Dear authors,

I have revised carefully the new version of your manuscript where I could see how much the suggested changes from the reviewers have improved your work.

In this regard, I am happy to say that I now consider that the manuscript follows the standards required to be published in PeerJ.

Congratulations!
Best regards,
Graciela Piñeiro